# Multi-Object Pedestrian Tracking Using Improved YOLOv8 and OC-SORT

**DOI:** 10.3390/s23208439

**Published:** 2023-10-13

**Authors:** Xin Xiao, Xinlong Feng

**Affiliations:** College of Mathematics and Systems Science, Xinjiang University, Urumqi 830046, China; 18229962268@163.com

**Keywords:** multi-object pedestrian tracking, YOLOv8, GhostNet, OC-SORT, object detection

## Abstract

Multi-object pedestrian tracking plays a crucial role in autonomous driving systems, enabling accurate perception of the surrounding environment. In this paper, we propose a comprehensive approach for pedestrian tracking, combining the improved YOLOv8 object detection algorithm with the OC-SORT tracking algorithm. First, we train the improved YOLOv8 model on the Crowdhuman dataset for accurate pedestrian detection. The integration of advanced techniques such as softNMS, GhostConv, and C3Ghost Modules results in a remarkable precision increase of 3.38% and an mAP@0.5:0.95 increase of 3.07%. Furthermore, we achieve a significant reduction of 39.98% in parameters, leading to a 37.1% reduction in model size. These improvements contribute to more efficient and lightweight pedestrian detection. Next, we apply our enhanced YOLOv8 model for pedestrian tracking on the MOT17 and MOT20 datasets. On the MOT17 dataset, we achieve outstanding results with the highest HOTA score reaching 49.92% and the highest MOTA score reaching 56.55%. Similarly, on the MOT20 dataset, our approach demonstrates exceptional performance, achieving a peak HOTA score of 48.326% and a peak MOTA score of 61.077%. These results validate the effectiveness of our approach in challenging real-world tracking scenarios.

## 1. Introduction

In recent years, the surge in traffic accident fatalities has been partly attributed to the growing vehicle populace. In response, considerable endeavors have been directed towards advancing pedestrian detection [1] and crosswalk tracking systems [2]. Multiple-Object Tracking (MOT), a computer vision task that identifies objects in videos and assigns distinct identities [3], has gained prominence. The advent of You Only Look Once (YOLO) in 2016, combined with the DeepSORT tracking algorithm, has found diverse applications in industries, agriculture, transportation, and beyond [4]. However, the realm of MOT is accompanied by several challenges. These encompass object occlusion and intersection, varying lighting conditions, a proliferation of targets, resembling appearances, and camera motion. In intricate scenes, objects might be concealed by occluders or other entities, impeding precise tracking. Changes in illumination can alter object characteristics, adversely influencing tracker performance. Scenarios housing numerous targets can impact tracking speed and precision. Similar object appearances can lead to confusion and tracking discrepancies. Lastly, camera movement can alter target views and positions, thereby affecting tracking accuracy.

Visual target tracking has been a dynamic research focus over the past decade. Initial classical approaches such as Meanshift [5] and particle filters [6] are primarily tailored for single-target tracking. However, their precision is modest, posing challenges in addressing intricate scenes. Nevertheless, the realm has witnessed rapid strides in deep learning, substantially elevating target detection performance and fostering the rise of detection-based tracking (DBT) methods. In DBT, objects are detected in each frame and then associated based on estimated instance similarity. Effective object detection leads to strong spatial proximity cues between objects across consecutive frames as gauged by metrics such as Intersection over Unions (IoUs) or center distances. However, while this approach thrives in simpler scenarios, it can falter in crowded or occluded environments. The drive toward heightened accuracy has prompted the development of more intricate and comprehensive networks in object recognition and tracking [7]. Nonetheless, it is important to note that greater accuracy does not always correlate with improved efficiency, particularly in terms of scalability and speed. In many industrial applications, such as automatic transmission, object counting, and video surveillance, the practical deployment of complex network models is hindered by factors such as cost constraints and the constrained availability of suitable chips. This scenario poses challenges in efficiently implementing these models on smaller processors with limited resources for real-time detection.

This paper propose a YOLOv8-OCSORT network model with improved accuracy and complexity balance for pedestrian target tracking. The contributions of this paper are summarized as follows:SoftNMS for improved pedestrian detection. In pedestrian detection, occlusion poses a common challenge, and traditional NMS techniques often result in missed detections. To address this issue, we introduce SoftNMS to enhance the performance of pedestrian detection under occlusion conditions. SoftNMS effectively handles overlapping bounding boxes and improves the accuracy of pedestrian detection.GhostNet for optimized model complexity. Traditional deep neural network models are typically complex and challenging to deploy on resource-constrained devices. In this study, we leverage a GhostNet module to optimize the YOLOv8 architecture. By sharing weights across multiple convolutional layers, it reduces model complexity while maintaining performance, enabling efficient execution on resource-limited devices.Integration of OCSORT tracking algorithm and REID model. We combine the OCSORT (GIOU) tracking algorithm with a mobileNetV2-based REID model. The OCSORT algorithm effectively handles occlusion, while the REID model ensures robust identity verification and tracking consistency. By integrating object detection and object tracking, our approach achieves outstanding performance in pedestrian tracking in complex scenarios.

## 2. Related Works

Multi-object tracking algorithms focus on tracing the trajectories of objects of interest, such as people and vehicles, across various frames in a video sequence [8]. Recent trends in multi-object tracking have been extensively reviewed by Guo et al. [9]. In their survey, the authors categorize these methods into three core groups: tracking-by-detection, joint detection and tracking, and transformer-based tracking. The tracking-by-detection framework is more commonly employed in industrial applications. In the tracking-by-detection paradigm, the MOT algorithm initially performs target detection on each frame of the video sequence. It subsequently refines the targets based on the context of the frame, effectively capturing all targets within the image. The process then transforms into an object association challenge between consecutive frames. This is achieved by constructing a similarity matrix using metrics such as the Intersection over Union (IOU) and appearance, followed by solving through algorithms such as the Hungarian algorithm or greedy algorithm.

The integration of Convolutional Neural Networks (CNNs) [10] has played a pivotal role in advancing detection algorithms, enhancing both accuracy and speed [11]. This progress has paved the way for the seamless integration of robust detection algorithms into target-oriented tracking systems, culminating in heightened tracking performance. RetinaNet, a notable object detector, harnesses CNNs alongside Feature Pyramid Networks (FPN) [12] and two Fully Convolutional Networks (FCNs) [13]. It effectively employs the focal loss function to assign greater weightage to challenging samples, skillfully balancing positive and negative samples.

YOLO detectors have evolved from V1 to V8, with the latest iterations achieving an optimal equilibrium between accuracy and speed. This evolution has positioned YOLO as a favored choice in numerous research endeavors [14]. The YOLO series provides efficient object detection with real-time processing capabilities, making it an appealing option for integration into tracking systems. The SORT algorithm leverages a simple Kalman filter for frame-by-frame data relevance and the Hungarian algorithm for association measurement. This simplicity empowers SORT to achieve commendable performance at high frame rates. However, due to its disregard for the object’s surface features, SORT’s accuracy is compromised when object state estimation uncertainty is high. Consequently, the introduction of cascade matching and other enhancements has given rise to DeepSORT, which exhibits superior performance on the foundation of SORT’s simplicity.

Joint detection and tracking algorithms commonly operate by detecting two consecutive frames in a video and subsequently employing diverse strategies to evaluate the similarity between targets in both frames. This facilitates simultaneous tracking and prediction. Prominent algorithms in this category include FairMOT [15], CenterTrack [16], and QDTrack [17]. On the other hand, transformer-based tracking integrates the Transformer architecture into multi-object tracking. Presently, two principal approaches stand out: TransTrack [18] and TrackFormer [19]. In TransTrack, the feature map of the current frame serves as the Key, while the Query is a composition of the target feature from the previous frame and a collection of target features acquired from the current frame. These inputs drive the entire network’s operation.

## 3. Proposed Method

Our proposed method consists of two main components: Enhanced YOLOv8 with softNMS and Ghost modules, and OC-SORT with GIOU for multi-object tracking.

### 3.1. The Proposed Network Structure

The YOLOv8 detection algorithm represents a notable progression within the YOLO series, integrating cutting-edge techniques and design principles to accomplish precise and efficient object detection. YOLOv8 builds upon the foundational architecture of YOLOv5 while introducing significant enhancements. The C3 module of YOLOv5 is substituted by the C2f module, drawing inspiration from the Cross-Stage Partial (CSP) concept. This amalgamation harnesses the strengths of the C3 module and the Efficient Lightweight Attention Network (ELAN) from YOLOv7, yielding refined gradient flow insights and a lightweight configuration. The YOLOv8 backbone adopts a Spatial Pyramid Pooling Fusion (SPPF) module, employing three successive max-pooling layers with a size of 5 × 5. These pooled feature maps are then concatenated, effectively encompassing objects of diverse scales. This blueprint ensures accurate detection prowess while upholding computational efficiency. In the neck component, the Path Aggregation Network and Feature Pyramid Network (PAN-FPN) methodology is embraced for feature fusion. This approach optimizes the integration and utilization of feature layers across varying scales, consequently enhancing overall detection performance. The neck module seamlessly integrates two upsampling operations, multiple C2f modules, and a decoupled head structure inspired by YOLOx. This combination accentuates object localization and classification precision.

To further improve the performance of YOLOv8, an improved YOLOv8 algorithm is proposed which combines the soft Non-Maximum Suppression (softNMS), GhostConv, and C3Ghost Modules. This combination aims to enhance the spatial awareness, suppress redundant detections, and optimize the network architecture. The network architecture of the proposed Network Structure is depicted in Figure 1.

#### 3.1.1. Softnms Implementation

In the original YOLOv8 framework, Non-Maximum Suppression (NMS) is employed to refine candidate boxes. However, the selection of the NMS threshold significantly influences the accuracy of pedestrian detection. A threshold set too conservatively may suppress valid positive instances, while an excessively lenient threshold could contribute to a rise in false positive instances. In light of the prevalent challenge of occlusion in pedestrian detection, conventional NMS often results in missed detections. To overcome this limitation, we integrated SoftNMS [20] to enhance pedestrian detection performance in occluded scenarios. SoftNMS is strategically tailored to address occlusion cases, and is mathematically expressed as follows:(1)$$\begin{array}{cc} \; & s_{i}=\left\{\begin{array}{c} s_{i}\left(1-IOU\left(M,t_{i}\right)\right),IOU\left(M,t_{i}\right)\geq N_{t},\\ s_{i},IOU\left(M,t_{i}\right)<N_{t}. \end{array}\right. \end{array}$$
where $s_{i}$ denotes the score of the *i*-th candidate box, M and $t_{i}$ denote the coordinates of the candidate box with the highest score and the coordinates of the *i*-th candidate box, respectively, the function *IOU*(.) quantifies the intersection over union ratio between the *i*-th candidate box and M, and $N_{t}$ designates a predetermined threshold.

#### 3.1.2. GhostNet Module Utilization

GhostNet stands out as a neural network architecture that strategically balances heightened accuracy with minimal computational overhead. It specifically addresses the limitations inherent in conventional deep neural network models, which tend to be excessively intricate for deployment on resource-limited devices. In this study, we harness the potential of the GhostNet module to amplify the performance of the YOLOv8 framework within scenarios constrained by resources. Our strategy involves optimizing the YOLOv8 architecture by replacing the original C2f module with a C3Ghost module sourced from GhostNet. By integrating the C3Ghost module, computational complexity is diminished through weight sharing across multiple convolution layers. Consequently, this adaptation culminates in a model that boasts fewer parameters without compromising on accuracy. This refinement is particularly advantageous for real-time applications such as pedestrian detection. Additionally, we replace the conventional conv module with the Ghostconv module, a streamlined alternative to traditional convolutional layers. This transition curtails the count of model parameters, thereby enabling the effective allocation of computational resources. Refer to Figure 2 for a visual representation of the GhostNet module.

In our study, we utilize the notation $X\in R^{c\times h\times w}$ to represent the input feature map, where *c* denotes the number of channels and *h* and *w* respectively represent the height and width of the feature map. The conventional convolution is defined as follows:(2)Y=X∗f+b.

Equation (Equation 2), $X\in R^{c\times h^{\prime }\times w^{\prime }}$ signifies a feature map with *n* output channels, where $h^{\prime }$ and $w^{\prime }$ respectively correspond to the height and width of the output feature map. The convolution operation is denoted by *, with a kernel size of k∗k, while *b* represents the bias term. The computational complexity of regular convolution, excluding the bias term, is approximately equal to $h\times w\times c\times n\times w^{\prime }\times h^{\prime }$. In the network’s shallower layers, $h^{\prime }$ and $w^{\prime }$ are larger, whereas in deeper layers *n* and *c* possess greater values. Motivated by this observation, the concept of Ghost convolution was introduced. It consists of two components: a conventional convolutional kernel that yields a limited number of feature maps, and the creation of surplus feature maps in a lightweight linear transformation layer, and can be expressed as follows:(3)$$Y^{\prime }=X*f^{\prime }+b.$$

Equation (Equation 3) represents a conventional convolutional layer that outputs a small number of feature maps, where $Y\in R^{h^{\prime }\times w^{\prime }\times m}$ represents the output feature and $f\in R^{c\times k\times k\times m}$ represents the size of this convolutional kernel. The number of channels of the output feature map is smaller than that of the conventional convolutional layer, i.e., m<n.
(4)$$Y_{ij}=\phi_{ij}\left(y_{i}^{\prime }\right).$$

Furthermore, Equation (Equation 4) represents the linear transformation layer that generates surplus feature maps. Here, $y_{i}$ signifies the *m* feature maps of $Y^{\prime }$. Each feature map within $Y^{\prime }$ undergoes a lightweight linear transformation $\phi_{ij}\left(j=1,2,...,s\right)$, yielding *s* feature maps. If convolution serves as the linear transformation, the final transformation is set as a constant transformation, resulting in *m* feature maps after transformation of m×(s−1) feature maps. The cumulative computation with Ghost convolution equates to $\left(s-1\right)\times m\times h^{\prime }\times w^{\prime }\times k\times k$. Refer to Figure 3 for a comprehensive depiction of YOLOv8 with GhostNet’s intricate structure.

#### 3.1.3. GIOU Loss Function

The Intersection over Union (IOU) distance is a widely adopted metric for evaluating the proximity between prediction and detection boxes within the detection space. This measure accurately characterizes the extent of their overlap while being independent of frame scaling. However, the IOU value remains static at 0 when no overlap exists between the boxes, which can lead to challenges in determining their correspondence. Moreover, even when the IOU values between trajectory prediction boxes and detection boxes are equal, their overlap positions might diverge significantly. This drawback is vividly illustrated in Figure 4, underscoring the insufficiency of the IOU distance in gauging the matching degree between detection and prediction frames.

As illustrated in Figure 4a,b, the IOU value remains consistent, while the overlapping positions between the detection frame and prediction frame exhibit notable dissimilarities. To mitigate this concern, the evaluation of frame intersection integrates position information. To this end, the Generalized Intersection over Union (GIOU) distance is introduced, which leverages the minimum bounding boxes of the detection and prediction frames to encapsulate their spatial positional relationship. The computation process of the GIOU distance is elucidated as follows:(5)$$GIOU=IOU-\frac{C-A\cup B}{C}$$
where *C* is the minimum frame area used to surround the detection frame and prediction frame, *A* is the area of the target trajectory prediction frame, and *B* is the area of the pedestrian detection frame.

Equation (Equation 5) highlights that the GIOU value remains invariable when the IOU equals 0, denoting an absence of intersection between the detection frame and prediction frame. A heightened value of parameter C, signifying increased separation between the detection and prediction frames, leads to a diminished GIOU value, and subsequently to an augmented GIOU distance. This phenomenon indicates a reduced level of correspondence between the frames. Conversely, a smaller GIOU distance indicates a higher degree of alignment between the frames.

#### 3.1.4. Evaluation Index

For the evaluation of the improved YOLOv8 algorithm, two main aspects were considered, namely, pedestrian detection and pedestrian tracking. The following evaluation indices were utilized to assess the performance of the algorithm in each aspect:Pedestrian Detection Evaluation: (a) Precision (P): denotes the ratio of correctly predicted positive detections to the total predicted positive detections. It gauges the algorithm’s capability to minimize false positives. (b) Recall (R): represents the ratio of correctly predicted positive detections to all actual positive instances in the ground truth. It assesses the algorithm’s effectiveness in minimizing false negatives. (c) Mean Average Precision (mAP): mAP serves as a widely utilized metric in object detection tasks. It computes the average precision across various object categories and IoU thresholds. This metric offers a comprehensive evaluation of the algorithm’s accuracy in object detection. The computation of these metrics is demonstrated as follows:
(6)$$P=\frac{TP}{TP+FP}$$
(7)$$R=\frac{TP}{TP+FN}$$
(8)$$AP=\int_{0}^{1}P\left(R\right)dR$$
(9)$$mAP=\frac{1}{N}\sum_{i=1}^{n}AP_{i}$$In these equations, TP (true positives) denotes the count of positive samples correctly predicted as positive, while FP (false negatives) corresponds to the number of positive samples erroneously predicted as negative. FP (false positives) represents the instances where negative samples are incorrectly predicted as positive. In this study, the total number of categories is set to 2. Moreover, the number of parameters, model size, and FLOPs serve as benchmarks for assessing the lightweight nature of a model. The quantity of parameters and model size primarily hinges on the network architecture. FLOPs, on the other hand, quantifies the computational complexity of the model by representing the number of calculations required for its operation.Pedestrian Tracking Evaluation: for object tracking, we used the CLEAR [22] evaluation indicator, which comprehensively considers FP, FN, and ID-Switch, and is a more common known as MOTA. CLEAR reflects the tracking quality of the tracker more comprehensively; however, as CLEAR ignores the ID characteristics of multiple targets, we additionally introduce IDF1 [23] to make up for the lack of MOTA in this regard. In addition, HOTA [24] is an indicator that has been proposed in recent years; it can reflect the effects of detection and matching in a balanced manner.

## 4. Experiments

In this section, we assess the performance of the improved YOLOv8 algorithm by conducting experiments on various datasets. Through comparisons with the baseline YOLOv8 and real-world experiments, we verify the algorithm’s effectiveness in improving object detection accuracy and enhancing tracking performance.

### 4.1. Datasets

The improved detector is trained on the CrowdHuman dataset [25]. The CrowdHuman dataset has a relatively large amount of data, with 15,000 images in the training set, 5000 in the testing set, and 4370 in the validation set. The proposed algorithm was evaluated on the MOT17 [26] and MOT20 [27] benchmark dataset. MOT17 has a total of 14 video sequences, of which seven sequences were used for training, with a total of 5316 frames, and seven sequences were used for testing, with a total of 5919 frames. MOT20 was set up for highly crowded challenging scenes, with four sequences and 8931 frames used for training and four sequences and 4479 frames for testing.

### 4.2. Experimental Settings

The experimentation was carried out on a high-performance computing setup encompassing an Intel(R) Xeon(R) Platinum 8255C CPU operating at 2.50 GHz and equipped with twelve vCPUs. An NVIDIA GeForce RTX 3080 GPU endowed with 10GB of VRAM, was employed. The system boasted a 40GB RAM configuration. These experiments were conducted on an Ubuntu 18.04 operating system, utilizing PyTorch 1.7.0 as the chosen deep learning framework, and GPU acceleration was facilitated through CUDA version 11.0. Python 3.8 served as the programming language. The training process incorporated the SGD optimizer with an initial learning rate of 0.01, which remained constant throughout training. A batch size of 16 was adopted for training, extending across 300 epochs. To avert overfitting, an early stopping strategy was implemented with a patience of 50 epochs.

### 4.3. Experimental Settings

#### 4.3.1. Pedestrian Detection Result

The training results for the improved YOLOv8 model are presented in Figure 5, showcasing the changes in metrics such as train loss, val loss, precision, recall, mAP@0.5, and mAP@0.5:0.95 as the number of epochs increases. The training loss and validation loss decrease over time, indicating improved model learning. Precision, recall, mAP@0.5, and mAP@0.5:0.95 exhibit an upward trend, showcasing the model’s enhanced detection performance. These results demonstrate the effectiveness of the pedestrian detection model in accurately identifying pedestrians in various scenarios.

As shown in Table 1, we evaluated our proposed model against the baseline YOLOv8 network. The integration of the SoftNMS technique yielded significant improvements across various performance metrics. Notably, we observed a 0.93% increase in precision, a 1.55% increase in recall, a 0.61% increase in mAP@0.5, and a remarkable 10.17% increase in mAP@0.5:0.95. Furthermore, the incorporation of GhostNet resulted in notable reductions in model complexity. We achieved a reduction of 39.98% in the number of parameters, corresponding to a 37.1% decrease in model size. Additionally, the FLOPs were reduced by 35.8%. The combined utilization of SoftNMS and GhostNet led to 3.38% increase in precision and 3.07% increase in mAP@0.5:0.95. These findings underscore the effectiveness of these improvement techniques in optimizing the YOLOv8n model, enabling enhanced object detection capabilities while maintaining a balance between accuracy and lightweight design.

Despite the individual performance degradation of the Ghost convolution, the combination of SoftNMS and Ghost convolutions results in improved performance compared to the baseline. This is due to the complementary nature of the two techniques. SoftNMS suppresses duplicate detections, compensating for the slight performance decrease caused by Ghost convolution. Moreover, the integration of Ghost convolution provides benefits such as reduced complexity and computation cost, enhanced feature representation, and increased receptive field, contributing to the overall improved performance.

Figure 6 illustrates the detection results of YOLOv8 and the improved models on sample pedestrian images. The images showcase various scenarios, including crowded streets and pedestrian crossings. The bounding boxes, along with the corresponding class labels and confidence scores, indicate the detected pedestrians and their associated certainty levels. The improved models demonstrate the ability to detect a higher number of pedestrians, including those in densely crowded areas. The bounding boxes accurately localize the pedestrians, even in challenging scenarios where individuals are closely packed together. Furthermore, the improved models show improved sensitivity in detecting pedestrians at various scales. They can successfully detect both small and large pedestrians, allowing for comprehensive coverage across different sizes and distances. This capability is particularly important in real-world scenarios where pedestrians may appear at varying scales.

In order to further validate the efficacy of the proposed model, we conducted a comparison of the visual effects using Grad-CAM. Grad-CAM is a technique that generates a heatmap representing the network’s attention to different regions of the input image. By applying Grad-CAM to both the original YOLOv8 and the improved model, we obtained the heatmaps illustrating their attention towards the target recognition, as depicted in Figure 7. The analysis of the heatmaps reveals that the improved model exhibits a higher intensity in the heatmap corresponding to the detection target area compared to the original YOLOv8. This observation suggests that the enhanced model is capable of extracting and leveraging the feature information of the detection target more effectively to a certain extent.

#### 4.3.2. Pedestrian Tracking Result

To assess the algorithm’s effectiveness, we utilized various detectors and trackors in the tracking process. Additionally, we utilized the ReID model mobilenetv2 for person re-identification.

For the ByteTrack tracker, the pedestrian tracking results on MOT17 are shown in Table 2. When combined with the public YOLOv8n detector, the MOTA was 33.599%, while the MOTP reached 81.432%. The IDF1 was 44.349%, while the HOTA score stood at 37.604%. Additionally, the FP (false positive) and FN (false negative) values were 3076 and 71,246, respectively. When the private YOLOv8n detector was used, the performance improved significantly. The MOTA increased to 50.529% and the IDF1 score rose to 57.377%, indicating improved detection and tracking accuracy. The HOTA score increased to 45.715%, showing enhanced overall performance. The FP and FN values decreased to 2222 and 53,026, respectively.

By integrating the SoftNMS technique into YOLOv8n, several metrics showed enhancements. The MOTP increased to 80.343%, indicating improved precision in object tracking. The IDF1 score improved to 57.473%, suggesting better detection and tracking capabilities. Additionally, the HOTA score reached 45.808%, demonstrating the effectiveness of the SoftNMS integration. The IDSW value decreased to 297, indicating a reduction in the number of identity switches and better tracking consistency. Incorporating GhostNet into the YOLOv8n architecture resulted in slightly lower MOTA of 43.759%, while maintaining a high MOTP of 80.602%; moreover, the FP value decreased to 1349, indicating a reduction in false positives. However, the IDSW value decreased to 233, indicating a reduction in the number of identity switches and better tracking consistency. The integration of both the SoftNMS and GhostNet techniques into the YOLOv8 algorithm led to a slight decrease in performance metrics. This can be attributed to the trade-off between detection accuracy and model complexity introduced by these techniques.

For the OCSORT tracker, similar performance trends were observed in Table 3. The combination with different YOLOv8n detectors consistently improved the tracking performance compared to the baseline YOLOv8. By integrating the private YOLOv8n detector into the OCSORT tracker, significant improvements were observed across multiple metrics. The MOTA increased to 56.546%, indicating enhanced tracking accuracy. The IDF1 score significantly improved to 62.324%, demonstrating enhanced detection and tracking capabilities. The HOTA score increased to 49.462%, indicating better overall tracking performance. Notably, the IDSW value decreased to 594, indicating a reduced number of identity switches. This yielded better tracking results compared to the baseline YOLOv8n detector. Incorporating the SoftNMS further improved the tracking performance. The MOTP improved to 80.013%. The IDF1 score reached 62.733%, indicating enhanced detection and tracking accuracy. The HOTA score increased to 49.889%, demonstrating the effectiveness of the combined approach. Moreover, the IDSW value decreased to 591, suggesting fewer identity switches, which is beneficial for maintaining consistent object identities throughout the tracking process.

When evaluating the OCSORT with GIOU tracker using different YOLOv8n detectors, as shown in Table 4, the overall performance was slightly better compared to using the original IOU.

Figure 8 depicts the variation of different performance metrics with respect to the alpha values, allowing us to identify optimal alpha values that strike a balance between these factors and make informed decisions for optimizing the multi-object tracking system.

The pedestrian tracking results on MOT20 dataset are shown in Table 5, Table 6 and Table 7. The ByteTrack tracker achieved an MOTA of 21.4% when combined with the Public YOLOv8n detector. However, this performance significantly improved to an MOTA of 57.436% when using the Private YOLOv8n detector. On the other hand, the OCSORT tracker demonstrated even higher performance, achieving an MOTA of 30.032% and 64.933% with the Public and Private YOLOv8n detectors, respectively. Similar to the findings on the MOT17 dataset, the OCSORT tracker with GIOU consistently outperformed the original OCSORT tracker across various metrics. These results highlight the effectiveness of using advanced trackers and improved detectors in pedestrian tracking tasks, emphasizing the importance of selecting appropriate configurations for achieving higher accuracy and reliability in tracking systems.

According to Table 8 and Table 9, compared to other methods, our approach demonstrates superior performance in terms of numerous metrics, including the MOTA, IDF1, MOTP, HOTA, and IDSW scores. These results validate the effectiveness of our method in achieving accurate and robust multiple object tracking.

#### 4.3.3. Tracking Effect Visualization

Figure 9 shows the tracking results of the proposed algorithm on real driving data. The results show the effectiveness of the proposed method in complex traffic scenarios.

## 5. Discussion

In summary, in this paper we conducted a comprehensive investigation into pedestrian detection and tracking by leveraging a fusion of YOLOv8-based methodologies and advanced techniques. The proposed enhanced YOLOv8 algorithm incorporating soft-NMS and Ghost modules exhibited notable enhancements in object detection performance. It achieved elevated precision, recall, and mAP when compared to the baseline YOLOv8 model. Our empirical assessment on the MOT17 and MOT20 datasets underscored the proposed algorithm’s effectiveness in detecting and tracking pedestrians across different demanding real-world contexts. The results underscored the superiority of the enhanced YOLOv8 algorithm in terms of both detection accuracy and speed, positioning it as a promising solution for real-time applications. Furthermore, the object tracking outcomes attained through the integrated framework of enhanced YOLOv8n, OC-SORT, and the MobileNetV2 model for REID showcased advancements in tracking accuracy, localization precision, and identity consistency. The lightweight and optimized characteristics of the proposed techniques coupled with their enhanced performance position them as viable options for deployment in resource-constrained environments and mobile applications. Our experimental findings not only validate the effectiveness of the proposed methods, they emphasize their potential to bolster pedestrian detection and tracking across diverse real-world scenarios.

Looking ahead, several promising avenues warrant exploration in future research. First, there exists an opportunity for further fine-tuning and optimization of the proposed algorithms to achieve even higher levels of detection and tracking performance. This refinement becomes especially crucial for addressing intricate and heavily occluded scenes. Second, expanding the applicability of the proposed techniques to object categories beyond pedestrians, such as vehicles or animals, can significantly broaden the utility of these methods across various domains. Third, evaluating the robustness and generalizability of the proposed algorithms on larger-scale datasets would provide valuable insights into their performance across diverse scenarios. In addition, exploring the integration of multi-modal data, such as depth information or thermal imaging, holds the potential to bolster detection and tracking accuracy, particularly in challenging environments. Furthermore, delving into real-time object tracking fused with high-level reasoning and decision-making algorithms could lead to more intelligent and context-aware tracking systems. Lastly, optimizing the proposed algorithms to align with hardware constraints and deployment contexts, such as embedded systems or edge devices, would enable their effective utilization in resource-constrained scenarios. By charting these future research directions, we can propel the field of pedestrian detection and tracking forward, resulting in more precise and efficient computer vision systems across domains such as surveillance, autonomous vehicles, and smart cities.

## Figures and Tables

**Figure 1 sensors-23-08439-f001:**
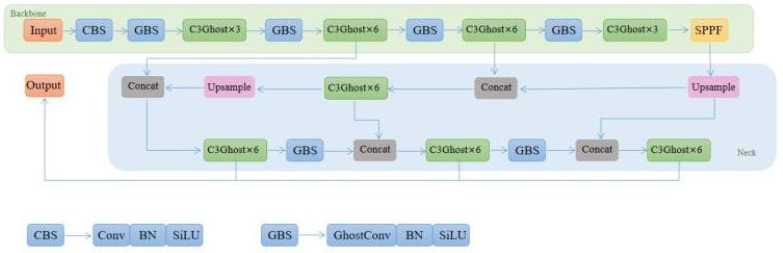
The structure of the Improved-YOLOv8.

**Figure 2 sensors-23-08439-f002:**
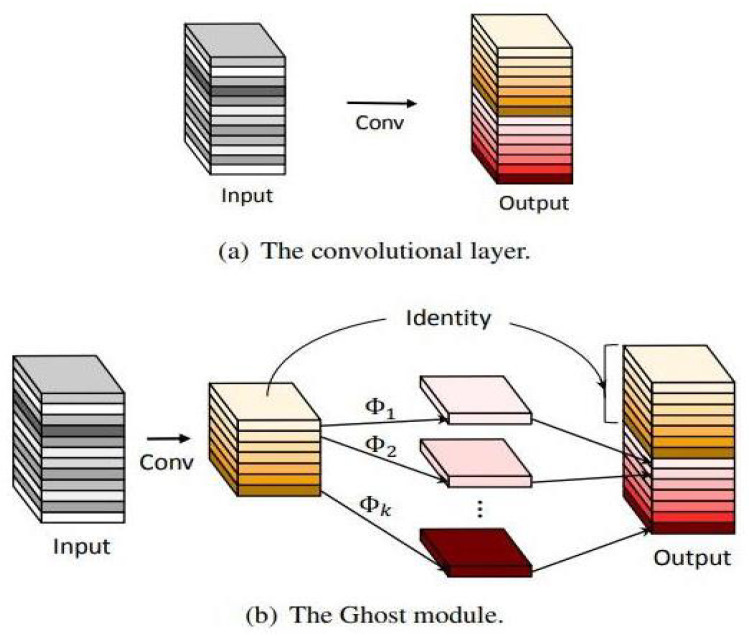
Schematic diagram of the convolutional layer and the GhostNet module [21].

**Figure 3 sensors-23-08439-f003:**
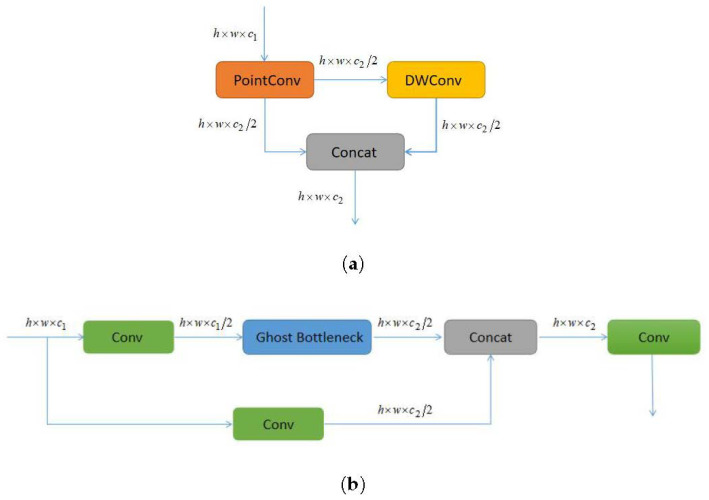
The structure of YOLOv8 with GhostNet: (**a**) Ghost module and (**b**) C3Ghost.

**Figure 4 sensors-23-08439-f004:**
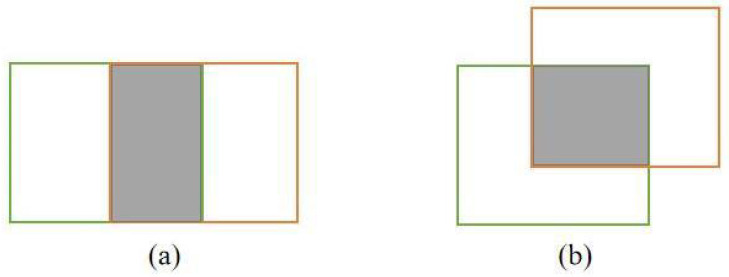
Coincident position relationship between detection frame and prediction frame: (**a**) horizontal overlap and (**b**) cross overlap.

**Figure 5 sensors-23-08439-f005:**
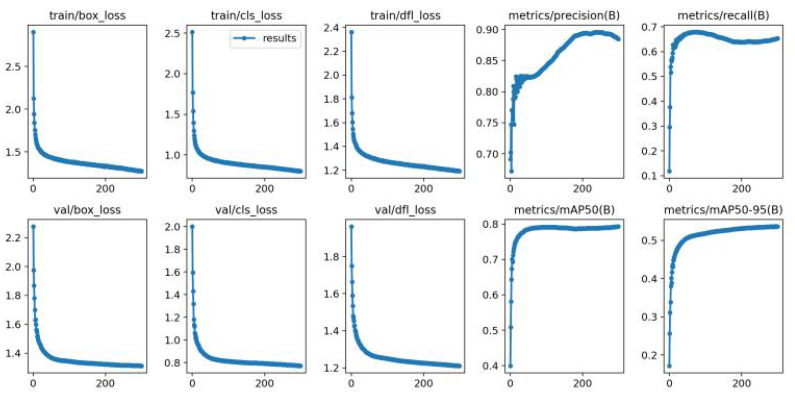
Training results of the the improved YOLOv8 model.

**Figure 6 sensors-23-08439-f006:**
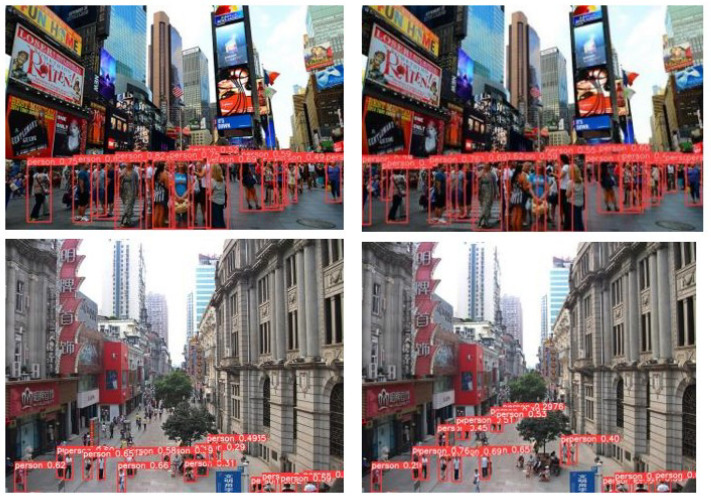
The detection results of the baseline model (**left**) and the improved model (**right**).

**Figure 7 sensors-23-08439-f007:**
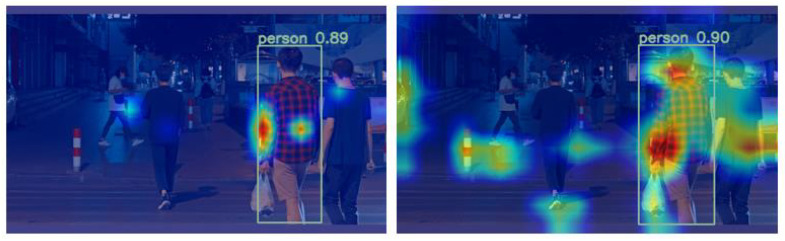
The heatmap of the baseline model (**left**) and the improved model (**right**).

**Figure 8 sensors-23-08439-f008:**
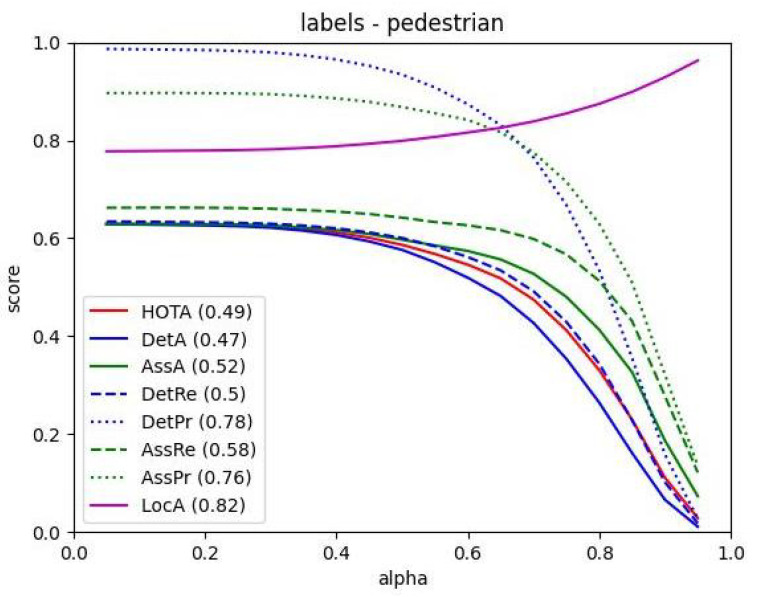
Graph of different performance indicators relative to alpha values.

**Figure 9 sensors-23-08439-f009:**
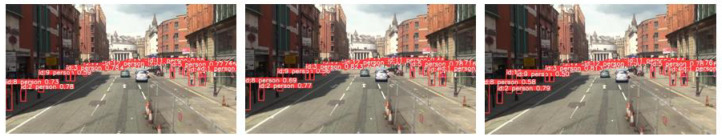
The pedestrian tracking effect of three consecutive frames in the video.

**Table 1 sensors-23-08439-t001:** Detection performance comparison.

Model	Precision	Recall	mAP0.5	mAP0.5:0.95	Parameters	Size/MB	FLOPs/G
YOLOv8n	0.857	0.710	0.820	0.521	3,005,843	6.2	8.1
+SoftNMS	0.865	**0.721**	**0.825**	**0.574**	3,005,843	6.2	8.1
+Ghost	0.841	0.682	0.789	0.467	**1,804,031**	**3.9**	**5.2**
+SoftNMS+Ghost	**0.886**	0.653	0.793	0.537	**1,804,031**	**3.9**	**5.2**

**Table 2 sensors-23-08439-t002:** Tracking performance comparison on MOT17 when using ByteTrack.

Detector	MOTA (↑)	MOTP (↑)	IDF1 (↑)	IDSW (↓)	HOTA (↑)	FP (↓)	FN (↓)	AssA (↑)	AssR (↑)
Pub_yolov8n	33.599	**81.432**	44.349	244	37.604	3076	71,246	47.195	50.919
Pvt_yolov8n	**50.529**	80.127	57.377	307	45.715	2222	**53,026**	50.391	55.168
+SN	50.248	80.343	**57.473**	297	**45.808**	2142	53,431	**50.668**	**55.668**
+GH	43.759	80.602	52.362	**233**	41.725	**1349**	61,575	48.763	52.636
+SN+GH	42.746	80.347	50.948	246	41.053	1567	62,481	47.964	52.268

Pub_yolov8n indicates public YOLOv8n, Pvt_yolov8n indicates private YOLOv8n, SN indicates SoftNMS, GH indicates Ghost.Same goes for the following.

**Table 3 sensors-23-08439-t003:** Tracking performance comparison on MOT17 when using OCSORT.

Detector	MOTA (↑)	MOTP (↑)	IDF1 (↑)	IDSW (↓)	HOTA (↑)	FP (↓)	FN (↓)	AssA (↑)	AssR (↑)
Pub_yolov8n	39.698	**81.278**	48.346	689	40.639	4889	62,139	45.895	50.669
Pvt_yolov8n	**56.546**	79.884	62.324	594	49.462	4046	**44,157**	52.255	57.625
+SN	56.224	80.013	**62.733**	591	**49.889**	4068	44,500	**53.096**	**58.286**
+GH	50.911	80.119	56.216	555	44.839	**2516**	52,054	48.285	53.351
+SN+GH	49.593	79.907	56.201	**490**	44.675	2852	53,263	48.787	54.081

**Table 4 sensors-23-08439-t004:** Tracking performance comparison on MOT17 when using OCSORT with GIOU.

Detector	MOTA (↑)	MOTP (↑)	IDF1 (↑)	IDSW (↓)	HOTA (↑)	FP (↓)	FN (↓)	AssA (↑)	AssR (↑)
Pub_yolov8n	39.698	**81.279**	48.392	686	40.644	4886	62,145	45.899	50.659
Pvt_yolov8n	**56.547**	79.884	62.278	592	49.471	4044	**44,160**	52.308	57.682
+SN	56.215	80.015	**62.823**	598	**49.915**	4068	44,503	**53.136**	**58.324**
+GH	50.907	80.119	56.127	557	44.794	**2516**	52,057	48.192	53.229
+SN+GH	49.592	79.906	56.615	**489**	44.857	2852	53,266	49.190	54.300

**Table 5 sensors-23-08439-t005:** Tracking performance comparison on MOT20 when using ByteTrack.

Detector	MOTA (↑)	MOTP (↑)	IDF1 (↑)	IDSW (↓)	HOTA (↑)	FP (↓)	FN (↓)	AssA (↑)	AssR (↑)
Pub_yolov8n	21.400	72.685	28.313	**1648**	20.495	7500	882,658	26.067	27.431
Pvt_yolov8n	**57.436**	78.622	**55.998**	2944	**42.254**	7691	**472,297**	**40.142**	**43.498**
+SN	53.543	78.901	53.613	2799	40.490	6792	517,516	39.287	42.305
+GH	45.576	78.746	46.547	2799	35.491	4594	610,104	35.339	37.921
+SN+GH	43.353	**78.937**	45.068	2652	34.334	**4330**	635,744	34.721	37.173

**Table 6 sensors-23-08439-t006:** Tracking performance comparison on MOT20 when using OCSORT.

Detector	MOTA (↑)	MOTP (↑)	IDF1 (↑)	IDSW (↓)	HOTA (↑)	FP (↓)	FN (↓)	AssA (↑)	AssR (↑)
Pub_yolov8n	30.032	72.559	36.127	4359	25.419	17,586	771924	28.226	30.146
Pvt_yolov8n	**64.933**	79.036	**64.315**	4208	**48.306**	14,361	**379,311**	**46.106**	**50.363**
+SN	61.077	**79.323**	61.731	**3761**	46.362	11720	426,143	44.939	49.040
+GH	54.595	79.115	54.547	4562	41.087	9511	501,098	39.501	43.403
+SN+GH	53.357	79.259	53.765	4287	40.570	**9237**	515,697	39.257	42.938

**Table 7 sensors-23-08439-t007:** Tracking performance comparison on MOT20 when using OCSORT with GIOU.

Detector	MOTA (↑)	MOTP (↑)	IDF1 (↑)	IDSW (↓)	HOTA (↑)	FP (↓)	FN (↓)	AssA (↑)	AssR (↑)
Pub_yolov8n	30.056	72.555	36.606	4250	25.625	17658	771,688	28.648	30.745
Pvt_yolov8n	**64.933**	79.035	**64.322**	4190	**48.326**	14352	**379,334**	**46.136**	**50.338**
+SN	61.077	**79.323**	61.824	**3757**	46.400	11711	426,160	45.003	49.029
+GH	54.595	79.115	54.557	4544	41.094	9508	501,120	39.506	43.365
+SN+GH	53.358	79.259	53.788	4269	40.632	**9229**	515,708	39.372	43.015

**Table 8 sensors-23-08439-t008:** Performance comparison with preceding SOTAs on MOT17.

Method	MOTA (↑)	MOTP (↑)	IDF1 (↑)	IDSW (↓)
CCC(2018) [28]	51.200	/	/	1851
GN(2020) [29]	50.200	/	47.000	5273
BLLSTM(2021) [30]	51.500	/	54.900	2566
STN(2021) [31]	50.000	76.300	51.000	3312
DET(2022) [32]	43.210	/	51.910	799
Pub_yolov8n	39.398	**81.278**	48.346	689
Ours	**56.215**	80.015	**62.823**	**598**

The ‘/’ character represents “not available” value.

**Table 9 sensors-23-08439-t009:** Performance comparison with preceding SOTAs on MOT20.

Method	MOTA (↑)	MOTP (↑)	IDF1 (↑)	IDSW (↓)
FairMOT(2021) [33]	61.800	/	**67.300**	5243
TransCenter(2021) [34]	62.300	**79.900**	50.300	4545
DET(2022) [32]	57.700	/	48.900	7303
MTrack(2022) [35]	63.500	/	69.200	6031
LADE(2022) [36]	54.900	79.100	59.100	**1630**
Pub_yolov8n	30.032	72.559	36.127	4359
Ours	**64.933**	79.035	64.322	4190

The ‘/’ character represents “not available” value.

## Data Availability

The data used in this study are publicly available.

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
