# Peer review of "Multi-Object Pedestrian Tracking Using Improved YOLOv8 and OC-SORT"

_sensors, 2023, doi:10.3390/s23208439_

Round 1

Reviewer 1 Report

In this paper, the authors propose an integrated approach that combines pedestrian detection using an improved YOLOv8 model with tracking using the OC-SORT tracking algorithm. According to the authors, this approach improves the accuracy of pedestrian detection, reduces the model's parameters and size, and achieves outstanding performance in challenging real-world tracking scenarios, as demonstrated on the MOT17 and MOT20 datasets.

The paper is well-written and the fundamental concepts are well explained.

Minor changes are necessary in the text:

Line 172: "The above equation" -> Please, explicitaly identify equations, figures and tables in the text.

Line 159: "Refer to Figure 2 for a visual representation of the Ghost module’s operation." -> The paper lacks more details explaining the figures. In several figures, the authors leave it to the reader to interpret what they intend to convey with the figure. Figures should be used to provide greater clarity to the reader, but the text should provide details about what the authors aim to illustrate with that figure. Please, check this to the other figures in the article.

Line 185: "IOU (Intersection over Union)". Acronyms should be spelled out in full the first time they appear. From the second occurrence onward in the text, it is no longer necessary to provide the full expansion. Another detail to be observed in the text is that it should have the full text followed by the acronym in parentheses, and not the other way around. For instance, Intersection over Union (IOU). Please check and correct this in the remaining acronyms that appear in the text.

Line 266: "Tables 1" -> "Table 1". Please, correct this for the remaining tables referenced in the text. 

Line 405: The text is exceeding the margin limits.

Tables 8 and 9: What does the '/' character in these tables mean? I suggest adding a caption to clarify its meaning for the reader.

I suggest to the authors to highlight in bold the values that indicate the best result. This way, it makes it easier for the reader to visualize the results.

I also suggest to the authors to improve the quality and readability of figures 1, 2, and 3. There is white space around the figures that can still be utilized. When using a 100% zoom, it can be observed that the readability of the figures is compromised.

Reviewer 2 Report

1.     In section 3, the structure of C3 module and C2f module should be more elaborated, with at least a figure detailing their architecture.

2.     Use a higher resolution for Figure 2 and add a citation for the figure. 

3.     Figure 3 misses label (a) and label (b)

4.     In section 4.3.1, explain why the softNMS+ghost leads to better performance even though with ghost convolution along the performance is worse than baseline.

5.     In Table 1, please also evaluate the inference speed for the models.

6.     In Figure, the author claims that the improved model with ghost conv is better at extracting detection targets. However, in Table 1 when without softNMS, ghost conv is worse than regular conv blocks. Please explain the discrepancy.

7.     Column 1 in tables 2, 3, 4, 5, 6, 7 should be better laid out.

8.     Please elaborate on whether GIOU is used with bytetrack. Moreover, the author could also try integrating bytetrack into OC_sort.

9.     Please also include public YoloV8n+softNMS in tracking result comparison

The overall quality of English and presentation are satisfactory.

Round 2

Reviewer 2 Report

All my previous concerns have been properly addressed in the current form of the manuscript

The presentation of the manuscript is satisfactory